# Plasma Spectroscopy of Various Types of Gypsum: An Ideal Terrestrial Analogue

**Abhishek K. Rai [1], Jayanta K. Pati [1,2], Christian G. Parigger [3,\*] and Awadhesh K. Rai [4]**

[1]  Department of Earth and Planetary Sciences, Nehru Science Centre, University of Allahabad, Prayagraj 211002, India
[2]  National Center of Experimental Mineralogy and Petrology, 14, Chatham Lines, University of Allahabad, Prayagraj 211002, India
[3]  Department of Physics and Astronomy, University of Tennessee, University of Tennessee Space Institute, Center for Laser Applications, 411 B. H. Goethert Parkway, Tullahoma, TN 37388, USA
[4]  Department of Physics, University of Allahabad, Prayagraj 211002, India
\*  Correspondence: cparigge@tennessee.edu; Tel.: +1-931-841-5690

**Abstract:** The first detection of gypsum ($CaSO_4 \cdot 2H_2O$) by the Mars Science Laboratory (MSL) rover Curiosity in the Gale Crater, Mars created a profound impact on planetary science and exploration. The unique capability of plasma spectroscopy, which involves in situ elemental analysis in extraterrestrial environments, suggests the presence of water in the red planet based on phase characterization and provides a clue to Martian paleoclimate. The key to gypsum as an ideal paleoclimate proxy lies in its textural variants and terrestrial gypsum samples from varied locations and textural types have been analyzed with laser-induced breakdown spectroscopy (LIBS) in this study. Petrographic, sub-microscopic, and powder X-ray diffraction characterizations confirm the presence of gypsum (hydrated calcium sulphate; $CaSO_4 \cdot 2H_2O$), bassanite (semi-hydrated calcium sulphate; $CaSO_4 \cdot \frac{1}{2}H_2O$), and anhydrite (anhydrous calcium sulphate; CaSO4), along with accessory phases (quartz and jarosite). The principal component analysis of LIBS spectra from texturally varied gypsums can be differentiated from one another due to the chemical variability in their elemental concentrations. The concentration of gypsum is determined from the partial least-square regressions model. The rapid characterization of gypsum samples with LIBS is expected to work well in extraterrestrial environments.

**Keywords:** laser-induced plasma; atomic spectroscopy; laser-induced breakdown spectroscopy; principal component analysis; partial least-square regression; gypsum; Mars

## 1. Introduction

Laser Induced Breakdown Spectroscopy (LIBS) is an atomic emission spectroscopic technique in which high power laser pulse is focused on the surface of the material and plasma is created. Light from the plasma is fed to the spectrometer to get spectral lines of the corresponding elements that are present in the material. The LIBS technique is applied in mineral and rock analyses in field and laboratory environments [1,2] for rapid measurements without involving sample preparation. It has a unique in-situ point detection capability. The LIBS technique was introduced in planetary exploration for the first time in 2012 when the Mars Science Laboratory (MSL) rover Curiosity studied the Gale Crater [3,4]. The ChemCam identified gypsum, which is a water-bearing non-metallic calcium sulphate mineral ($CaSO_4 \cdot 2H_2O$), as veins on the rim of Endeavour Crater, Mars as veins [5,6]. Consequently, it was suggested that gypsum on Mars formed due to the evaporation of a large lake [7–9] under a suitable physico-chemical environment. Sulphate salts are useful markers for decoding paleoenvironments as

they precipitate from aqueous fluids suggesting the presence of liquid water. Further, the aqueous geochemistry promotes the proliferation of microbial life and its consequent preservation in planetary environments [10].

Gypsum provides an ideal habitat for endolithic microbial communities, protection from harmful ultra-violet radiations, and supports their proliferation even under High Arctic conditions [11] to hot springs [12] environments with temperatures larger than 50 °C. Microbes thrive within gypsum crystals in terrestrial impact craters, such as Haughton structure [13–15], and the extraterrestrial landscapes are endowed with such structures [16]. Based on terrestrial evidences, the possible presence of microbial life is ptredicted in gypsum veins and in other sulfate minerals on Mars [17,18], other planets of the solar system, their moons, asteroids, and comets.

The MSL Curiosity rover characterized many calcium sulfate veins at Gale crater, Mars, including gypsum at Martian surface with nominal change in elemental composition [19]. Dunes of gypsum are also detected at the Martian North pole [20,21]. The presence of sulphates that are associated with clays at the base of the Mount Sharp by Compact Reconnaissance Imaging Spectrometer (CRISM) is also pointed out by Milliken et al. 2010 [22] and also by Opportunity rover at Meridiani Planum [23].

Various hypotheses explaining the origin of the massive Martian sulphate deposits exist, which advocate hydrothermal processes, oxidation of sulphides, evaporative deposition, and bedrock leaching by volcanic vapors [23–29].The report of gypsum veins from Mars, its formation in various terrestrial environments, and the suitability for the proliferation of life have made this hydrated sulfate mineral an ideal terrestrial analogue [30]. In addition, the texturally different types of gypsum, which in turn control the physical and chemical conditions of their formation, have attracted significant attention [31]. In the present work, gypsum samples from varied terrestrial environments and textural types have been studied for their inter-sample comparison while using LIBS data combined with multivariate statistical analysis methods. The data generated can be compared with the available ChemCam data on gypsum from Mars and the upcoming Chandrayaan-2 experiments from Moon [32].

## 2. Mineralogy and Geochemistry of Gypsum

Gypsum crystallizes in monoclinic system and its structure contains parallel layers of $(SO_4)^{-2}$ group bonded with $(Ca)^{+2}$. Sheets of $H_2O$-molecules presenting very weak bonds and breakdown on heating separate thee $CaSO_4$ molecules [33,34]. Calcium atoms are enclosed by two water molecules and six sulphate oxygen atoms, though the sulphur atoms are at the middle of sulphate oxygen tetrahedra. The water molecules have two oxygen atoms and calcium as their first neighbors. The arrangement consists of $SO_4$ tetrahedra, $CaO_8$ polyhedra, and weak hydrogen bonds made among the O atoms of nearest-neighbour $SO_4$ tetrahedra and intercalated $H_2O$ molecules [35]. The diaphaneity of gypsum varies from transparent to opaque and it is usually light in color. The habit of gypsum is platy, columnar, fibrous, needle-like, lenticular, forms massive aggregates, and twins-swallowtail. It is a relatively soft mineral (1.5–2: Mohs' Scale of Hardness) and it gets weathered easily. Gypsum constitutes 79.1% of calcium sulphate and 20.9% water by weight. Since gypsum is known to be formed by several processes (digenesis [36], chemical weathering [37], evaporation [38], hydrothermal activities, etc. [39]) and under varied physical and chemical environments, this has led to useful reconstruction of the paleoenvironment [40,41]. Sometimes, lamination occurring in gypsum (medium to coarse grains) from the evaporate basin can lead us to know about the changes in the basin i.e., water composition and water level. Gypsum laminas (like biolamina) crystallizes in the shallow part, approximately 200 m can be either deformed by cyclic droughts, like mud cracks, or ruptured by crystallizing sulphates. Gypsums crystallizing in the deeper zones constitute elongated crystals (nearly 20–30 cm long), which are distorted in one direction due to the low floor current activity, and they are represented as proper paleocurrent indicators. Selenite is one the varieties of gypsum, which forms under stable condition at few to several meters depth, and they can grow up to 10 m. Laminated selenites forming under deeper zones show ripple marks, turbidities, and slump structure with fragments of older and more lithified gypsum.

Calcium sulfate can crystallize in three phases when it comes with contact with water gypsum ($CaSO_4 \cdot 2H_2O$), hemihydrate or bassinite ($CaSO_4 \cdot \frac{1}{2}H_2O$), and anhydrite ($CaSO_4$). Gypsum may lose crystallization water molecules, becoming bassanite (hemihydrate: $CaSO_4 \cdot \frac{1}{2}H_2O$) or anhydrite ($CaSO_4$) under suitable geological and temperature conditions, pressure, and dissolved electrolyte [42,43]. The formation of gypsum takes place through a complex multi-step processes [43]. It can form in evaporite beds during evaporation of water near lake and sea water. Gypsum can diagenetically form by oxidation. It can form in association with sedimentary rocks, as the oxidation of existing sediments leading to gypsum formation and replaces the other minerals in the sediments [44]. It can also occur in hot spring from volcanic vapors, as well as in sulfate solutions in veins [19]. Hydrothermal anhydrite in veins is commonly hydrated to gypsum by groundwater in near-surface exposures. Gypsum can also occur in a flower-like form with embedded sand grains that are called desert rose in arid areas [45].

## 3. Materials and Methods

The five gypsum samples are collected by one of us (JKP) in person from different geographic locations and environments of formation. They are shown in Figure 1 and they include the following varieties: 1: "Desert Rose" Gypsum, Sahara Desert, Morocco (MDRG). 2: Acicular Gypsum, Morocco (MAG); 3: Platy Gypsum, Kachchh, Gujarat (KG); 4 Flaky Gypsum, Rajpura-Dariba Cu-Pb-Zn Mines, Rajasthan (RDG1); and, 5: Bladed Gypsum, Rajpura-Dariba Cu-Pb-Zn Mines, Rajasthan (RDG2). Figure 2 shows the SEM images of all five samples.

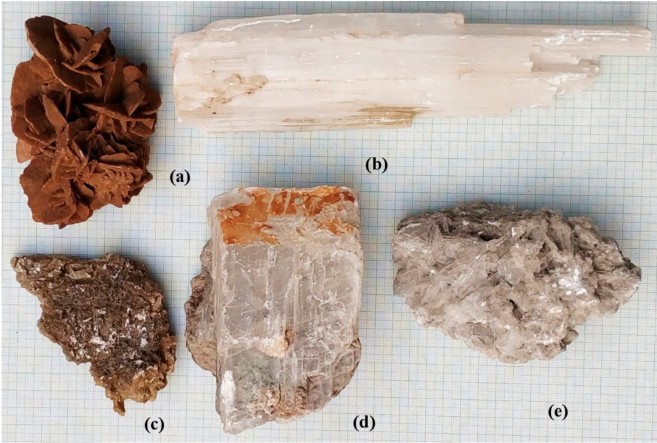

**Figure 1.** Specimens used in the present study and they include (**a**) Desert Rose Gypsum, Sahara Desert, Morocco (MDRG). (**b**) Acicular Gypsum, Morocco (MAG). (**c**) Gypsum, Kachchh, Gujarat, India (KG). (**d**) Gypsum, Rajpura-Dariba Cu-Pb-Zn Mine, Rajasthan, India (RDG1). (**e**) Gypsum, Rajpura-Dariba Cu-Pb-Zn Mine, Rajasthan, India (RDG2).

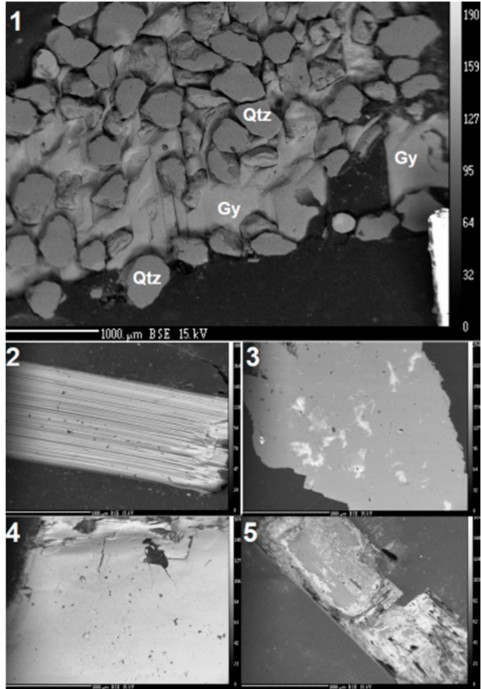

**Figure 2.** Electron microscope (SEM) images of the samples used in the present study which include: **1:** Desert Rose Gypsum, Sahara Desert, Morocco (MDRG), Gypsum, Gy, and quartz, Qtz, are identified. **2:** Acicular Gypsum, Morocco (MAG). **3:** Gypsum, Kachchh, Gujarat, India (KG). **4:** Gypsum, Rajpura-Dariba Cu-Pb-Zn Mine, Rajasthan, India (RDG1). **5:** Gypsum, Rajpura-Dariba Cu-Pb-Zn Mine, Rajasthan, India (RDG2).

### 3.1. Sample Description

Desert Rose sample (MDRG; Sahara, Morocco) comprise five rose-like formations of platy crystals. They form an array of circular nearly flat plates, where the plates are having sharper edges and relatively thick center. The diameter of biggest plate is about 5 cm and smallest is around 1.5 cm. The color is pinkish brown like sand (5YR 4/6; Munsell Color Chart). The sand encrustation on the top surface is about a few mm thick [45].

Acicular gypsum (MAG; Morocco from Ouarzazate Province) constitutes a column-like aggregate of needle-shaped gypsum crystals. The length of the gypsum is around 24 cm with a width around 3 cm. The color of the gypsum is white (5GY 9/2; Munsell Color Chart). The sample does not contain any visual impurity [46,47].

Kachchh gypsum (KG; Gujarat, India) is light brown color (7.5 YR 7/8; Munsell Color Chart) and contains randomly arranged sub-millimeter-thin flakes/plates. The effect of surficial weathering in the sample is evident. The length of the sample is about 10 cm and its width is 2 cm [48–51].

Rajpura-Dariba gypsum 1(RDG1; Dariba Mines, Rajasthan) is transparent and colorless. The sample is having sub-millimeter platy layers. The length of the sample is 11.7 cm, with width of 6 cm [52].

Rajpura-Dariba gypsum 2 (RDG2; Dariba Mines, Rajasthan, India) is off-white in color (5 Y 9/2; Munsell Color Chart). The sample is having blade like petal presentingno fixed orientation and the blades are very thin (around 0.5 cm). The sample length and width are 12.3 and 2.5 cm, respectively [52].

### 3.2. Powder X-ray Diffraction (XRD)

The powdered gypsum samples were characterized by X-ray Diffractometer (Philips Pan Analytical) while using CuKα radiation and Ni filter. The scans were made between 10 to 75°

2θ at a speed of 2° per minute, divergence slit = 1°, anti-scatter slit = 1°, receiving slit = 0.30 mm, generator voltage of 40 kV, and tube current of 30 mA.

### 3.3. Experimental Set-Up

The experimental setup consists of a frequency-doubled (532 nm) Nd:YAG laser (Continuum Surelite III-10). The plasma is created while using a 15 mJ laser pulse having a pulse width of 4ns with 10 pulses per second of repetition rate. A convex lens of focal length 15 cm is used to focus the laser on the sample to create hot plasma. The light that was emitted by the plasma at the sample surface was collected by a set of collection optics (CC52 collimator, Andor Technology, Belfast, United Kingdom) and focused into the optical fibre bundle. The optical fiber bundle that was delivered the light to the Mechelle spectrograph (ME5000, Andor Technology, USA) equipped with an intensified charge-coupled device (ICCD) (iStar 334, Andor Technology, USA). The spectral signatures of the atoms and ions form of the plasma are collected after a short interval of time to avoid the background continuum. Therefore, the gate delay and gate width are 0.7 μs and 4 μs, respectively. A detailed analysis of the spectral lines uses Andor SOLIS software and National Institute of Standards and Technology (NIST) atomic spectroscopic database [53].

### 3.4. Multivariate Analysis

Multivariate analysis is a tool that can be used to maximize the information that was extracted from a huge number of datasets, here in the present case spectral data of LIBS spectra of the samples. Single spectra of LIBS can contain more than 8000 data points, thus, it is very difficult to distinguish the spectral variables (wavelength, intensity) of the samples with chemical heterogeneity [54,55].

Multivariate analyses techniques, such as principal component analysis (PCA) and partial least squares regression (PSLR), are used in the present study. PCA is an unsupervised method that reduces the dimensionality (number of features) within a dataset, while still retaining as much information as possible. It is an orthogonal transformation that reduces the spectral data by projecting them into lower dimensions, called principal components [56]. For the computation of the first principal component, it is the linear combination of variables on which most data variation can be projected. Additionally, for the succeeding component, it is perpendicular to the previous principal component, so that it contains maximum variation. PLSR is one of the multivariate methods that are used in the interpretation of spectra (independent variables) to predict concentrations (dependent variables) of the element in the sample. For making the regression model, the concentration of calcium and sulfur is taken from the analysis of Electron Microprobe Analyzer [57].

## 4. Results and Discussion

### 4.1. X-ray Diffraction Analysis

X-ray diffraction studies on powered samples reveal the presence of gypsum and anhydrite in all samples, as illustrated in Figure 3, having a prominent d-spacing of 7.66 Å, 4.27 Å, 3.82 Å, 3.08 Å, and 2.69 Å 1.82 Å, respectively. The presence of Jarosite (d-values: 5.59 Å and 3.72 Å) is noticed in gypsum from Matanumadh, Kachchh, whereas in Desert Rose gypsum (Morocco) sample, quartz is observed with characteristic d-values of 3.36 Å (I/I$_0$: 59.42) and 1.38 Å (I/I$_0$: 5.59).

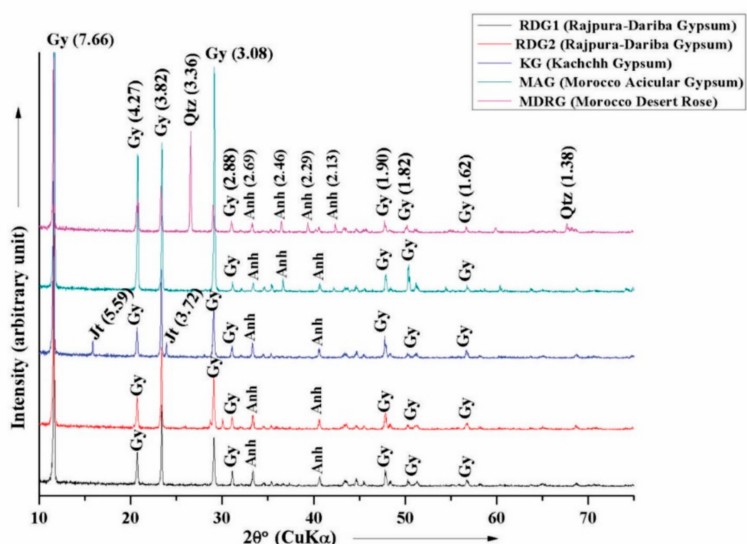

**Figure 3.** Powder X-ray diffraction (XRD) patterns of different gypsum samples showing the presence of gypsum (Gy; d = 7.66, 4.27, 3.82 and 3.08 Å), anhydrite (Anh; d = 2.69, 2.46 and 2.29 Å), quartz (Qtz; d = 3.36 Å), and jarosite (Jt; 5.59 and 3.72 Å).

## 4.2. LIBS Analysis

The LIBS spectra of different gypsum samples are recorded in the wavelength range between 200 and 800 nm. Figure 4 illustrates the results.

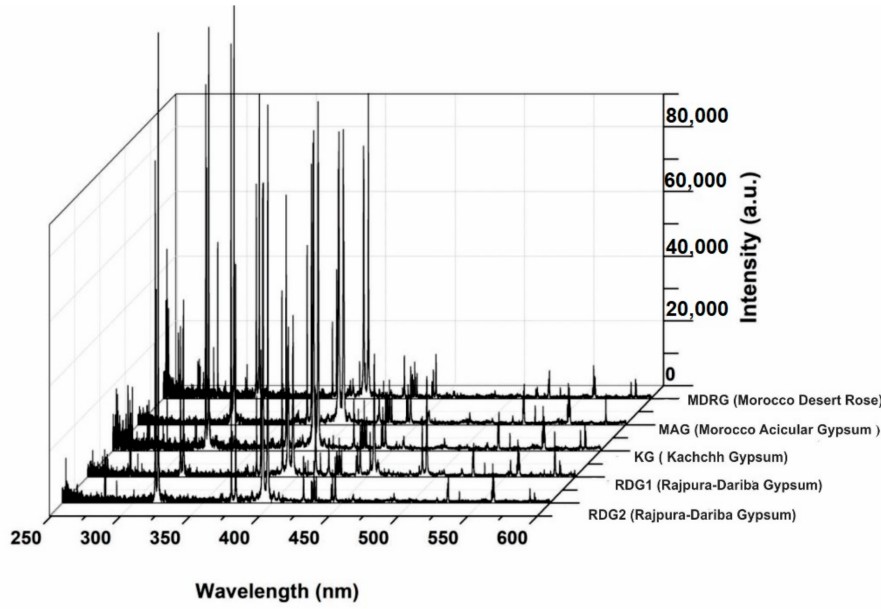

**Figure 4.** Typical laser-induced breakdown spectroscopy (LIBS) spectra (250–600 nm) of different gypsum samples analyzed during the present study stacked to show the relative intensity variations amongst the samples.

The differentiation of LIBS spectra of similar types of samples is difficult by visual inspection, but careful analysis of the spectra reveals differences in the intensity of the gypsum samples because of their variable elemental concentrations. The wavelengths of different spectral lines that were present in the LIBS spectra of all five gypsum samples have been identified while using the NIST atomic spectroscopic database [53]. Figures 5–8 display details of the spectra, together with the identified elements.

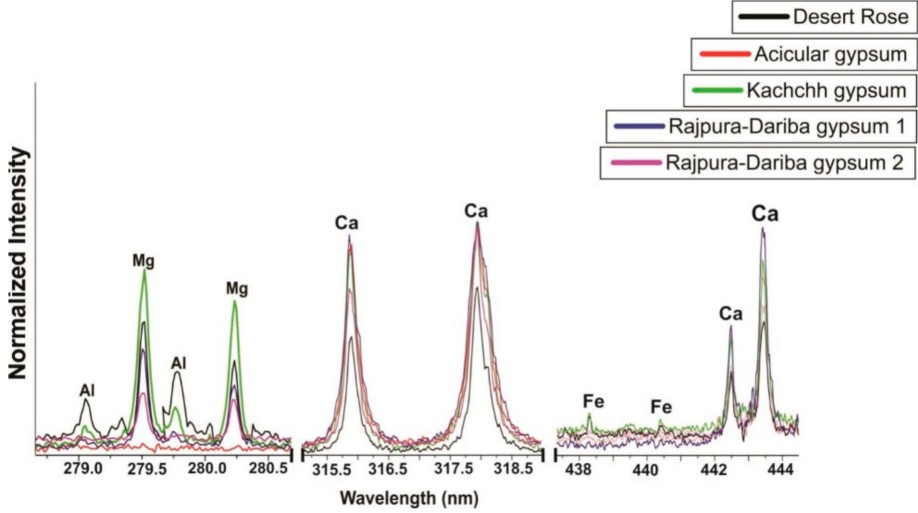

**Figure 5.** Spectral regions of recorded Al, Mg, Ca, and Fe data for the gypsum samples.

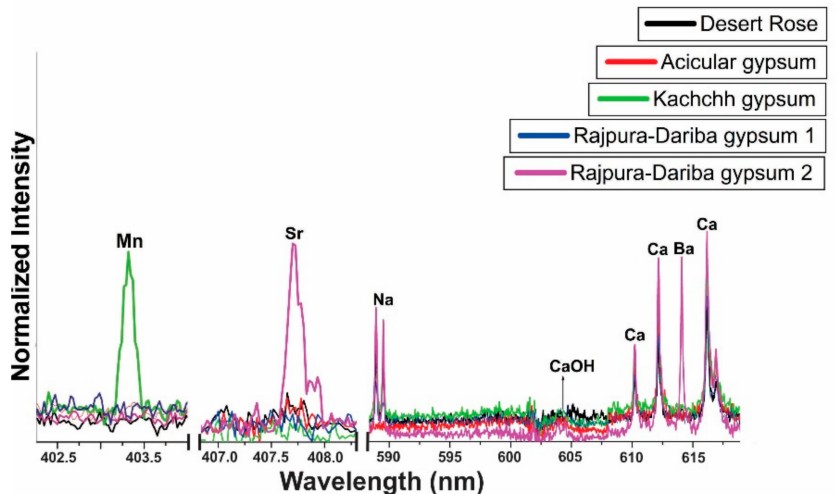

**Figure 6.** Spectral regions of recorded Mn, Sr, Ca, and Ba atomic and CaOH molecular data for the gypsum samples.

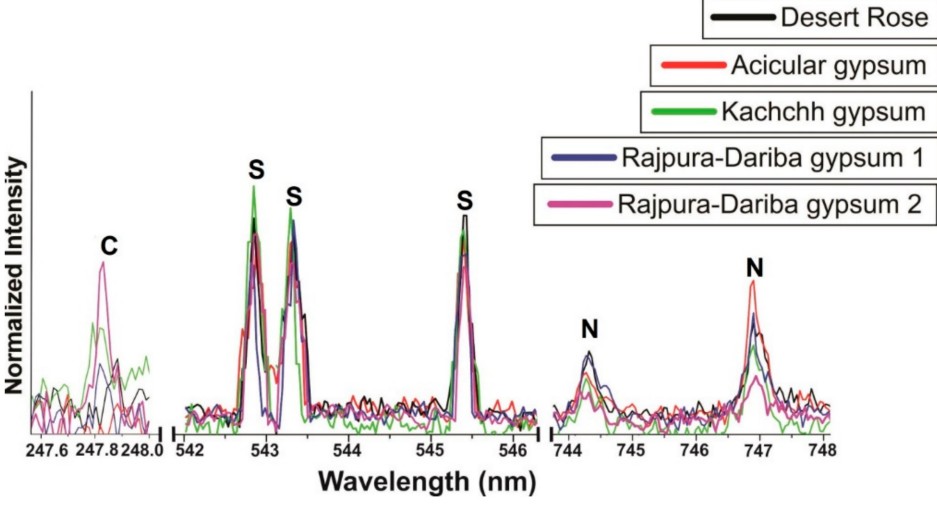

**Figure 7.** Spectral regions of recorded C, S, and N atomic data for the gypsum samples.

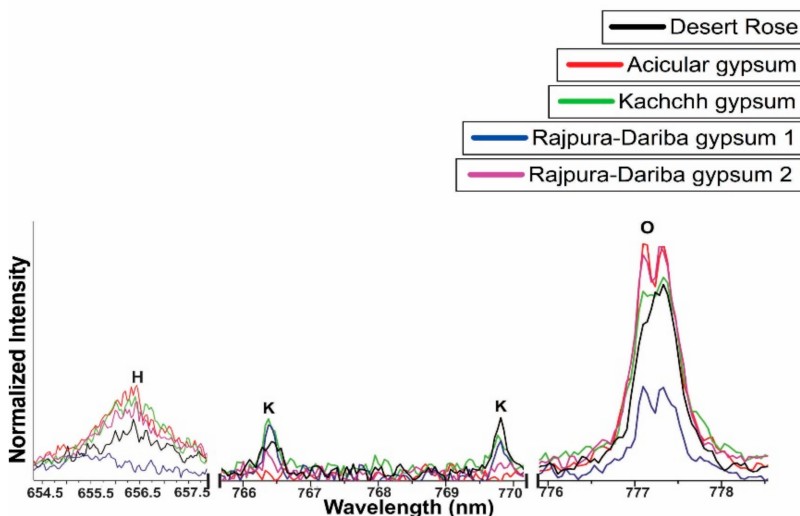

**Figure 8.** Spectral regions of recorded H, K, and O atomic data for the gypsum samples.

Figures 5–8 indicate that the intensities of spectral lines of the elements are different in the LIBS spectra of gypsums that formed in different conditions/environment. Thus, one can predict the environment and the process of formation with the information of elemental analysis or composition of gypsum, which cannot be achieved by their physical appearance. Persistent spectral lines, along with the other weak spectral lines of calcium, are observed in all five samples. Similarly, in the observation of Forni et al. [58,59], a molecular band near 606 nm that was probably due to presence of CaOH is also observed in the LIBS spectra of all five gypsum samples.

If the concentration of any element in the sample is very high, its molecular bands must be present in the laser induced plasma of the sample [60]. The presence of sulfur lines (542.8 nm, 543.2 nm, 545.3 nm) in the LIBS spectra of gypsum also confirms its presence, as indicated in Figure 8. Similarly, the presence of carbon line at 247.8 nm and oxygen at 777.1 nm indicates the presence of carbonates in the present study. Strong spectral lines of magnesium at the wavelengths 279.5 nm, 285.2 nm, 383.8 nm, etc. are present in four gypsum samples, except in the acicular sample (MAG). Magnesium occurs in the defect sites of the gypsum lattice, since they form large soluble complexes [61]. The intensity of the spectral line of magnesium varies, depending upon its concentration in the sample. The presence of prominent line of Si in MDRG is due to the presence of quartz grains from the sand, which bonds with moisture on the top of gypsum plates. The Ca lines are prominent in all gypsum samples, except MDRG, as quartz grains dominantly mask the analyzed spots. The presence of spectral lines of Fe in KG and MDRG are due to jarosite and quartz, respectively. Gypsum comprises abundant fluid phase, which is a mixture of various elements K, Ti, Mn, and Sr in minor to trace amounts. Their concentration in gypsum is used in stratigraphic correlation, paleosalinity estimation, and to decipher the origin of brine and diagenesis [62,63]. The presence of spectral lines of Mn and Fe in KG indicates that the terrigenic source deposits mainly siliciclastic sediments or clay minerals into the sedimentary basin. Sr lines are found in RDG2, as Sr can replace Ca in the gypsum lattice [64]. The presence of strontium in gypsum can take place in many ways, like digenetic process, bacterial sulphate reduction, or by dissolution or recrystallization. The abundance and scarcity of Sr content in gypsum indicate the episodic salinity fluctuation of the basin, and this also reveals the paleohydrology of the basin [65]. Similarly, the presence of spectral line of Ba (455.4 nm and 614.1 nm) may be correlated as a possible tracer of fluid circulation in hydrothermal water [66]. Tables 1 and 2 sumarize the wavelengths of the spectral lines that were observed in the LIBS spectra of all five gypsums. The same calcium lines are identified in all the samples and the wavelengths are listed below in Table 2.

**Table 1.** Spectral wavelengths of atoms observed in laser induced breakdown spectroscopic spectra of different gypsum samples: Hydrogen, nitrogen, oxygen, sodium, magnesium, aluminum, and silicon.

| Element | Acicular Gypsum | Desert Rose | Kachchh Gypsum | RajpuraDariba Gypsum 1 | RajpuraDariba Gypsum 2 |
|---|---|---|---|---|---|
| H (1) | 656.3 (I) | 656.3 (I) | 656.3 (I) | 656.3 (I) | 656.3 (I) |
| N (7) | 744.2(I), 746.8(I), 868.3(I) | 744.2(I), 746.8(I), 868.3(I) | 744.2(I), 746.8(I), 868.3(I) | 744.2(I), 746.8(I), 868.3(I) | 744.2(I), 746.8(I), 868.3(I) |
| O(8) | 777.4(I), 844.6(I), 926.6(I) | 777.4(I), 844.6(I), 926.6(I) | 777.4(I), 844.6(I), 926.6(I) | 777.4(I), 844.6(I), 926.6(I) | 777.4(I), 844.6(I), 926.6(I) |
| Na (11) | 588.9(I), 589.5(I) | 588.9(I), 589.5(I) | 588.9(I), 589.5(I) | 588.9(I), 589.5(I) | 588.9(I), 589.5(I) |
| Mg (12) | - | 279.0(II), 279.5(II), 280.2(II), 285.2(I), | 279.0(II), 279.5(II), 280.2(II), 285.2(I), | 279.0(II), 279.5(II), 280.2(II), 285.2(I), | 279.0(II), 279.5(II), 280.2(II), 285.2(I), |
| Al (13) | - | 308.2(I), 309.2(I), 394.3(I), 396.1(I) | 308.2(I), 309.2(I), 394.3(I), 396.1(I) | - | - |
| Si (14) | - | 220.7(I), 221.0(I), 221.6(I), 250.6(I), 251.4(I), 251.6(I), 251.9(I), 252.8(I), 288.1(I), 390.5(I), 413.1(II), 557.6(II), 634.7(II), 716.5(I), 741.5(I) | 220.7(I), 221.0(I), 221.6(I), 250.6(I), 251.4(I), 251.6(I), 251.9(I), 252.8(I), 288.1(I), 390.5(I), 413.1(II), 557.6(II), 634.7(II) | 252.8(I), 288.1(I) | - |

**Table 2.** Spectral wavelengths of atoms observed in laser induced breakdown spectroscopic spectra of different gypsum samples: Sulfur, potassium, titanium, manganese, iron, strontium, barium.

| Element | Acicular Gypsum | Desert Rose | Kachchh Gypsum | Rajpura-Dariba Gypsum 1 | Rajpura-Dariba Gypsum 2 |
|---|---|---|---|---|---|
| S (16) | 542.8(I), 543.2(I), 545.3(I) | 542.8(I), 543.2(I), 545.3(I) | 542.8(I), 543.2(I), 545.3(I) | 542.8(I), 543.2(I), 545.3(I) | 542.8(I), 543.2(I), 545.3(I) |
| K (19) | 766.4(I), 769.8(I) | 766.4(I), 769.8(I) | 766.4(I), 769.8(I) | 766.4(I), 769.8(I) | 766.4(I), 769.8(I) |
| Ti (22) | - | - | 334.9(II), 336.1(II) | - | - |
| Mn (25) | - | - | 257.6(II), 403.0(I) | - | - |
| Fe (26) | - | 238.2(II), 239.5(II), 240.4(II), 249.3(II), 252.3(I), 259.9(II), 258.5(II), 259.8(II), 261.1(II)263.1(II), 271.9(I), 273.9(II), 274.9(II), 275.5(II), 344.0(I), 358.1(I), 373.7(I) | 238.2(II), 239.5(II), 240.4(II), 249.3(II), 252.3(I), 273.9(II), 274.9(II), 275.5(II), 344.0(I) | - | - |
| Sr (38) | - | - | - | - | 407 (I) |
| Ba (56) | - | - | - | - | 455.4(II), 493.3(II) |

The hydration level in gypsum depends upon the hydrogen and oxygen content; calcium sulfate either occurs as non-hydrated (anhydrite $CaSO_4$) or hydrated phase (bassinite $CaSO_4 \cdot \frac{1}{2}H_2O$ and gypsum: $CaSO_4 \cdot 2H_2O$). The H alpha emission line at 656.5 nm is sensitive to hydrogen content, which can can differentiate the hydration levels of calcium sulfates in the samples. Similarly, the emission line of oxygen at 777.1 nm is sensitive to correlate the hydration level in the sample [67]. The spectral lines of both elements are detected, but the respective intensities of the spectral lines of hydrogen and oxygen in samples RDG-1 are very low. Thus, their concentration in RDG1 is also low, which establishes that the calcium sulfates observed may be bassanite.

The observed calcium atomic lines, Ca (20), are the same for all of the samples. The wavelengths of Ca I and Ca II are: 315.8(II), 317.9(II), 370.6(II), 373.6(II), 393.2(II), 396.7(II), 370.5(II), 422.6(I), 430.2(I), 442.5(I), 443.9(I), 445.4(I), 501.9(II), 518.8(I), 558.7(I), 610.2(I), 612.2(I), 616.2(I), 645.6(II), 646.2(I), 649.5(I), 647.2(I), and 720.1(I) nm.

### 4.3. Principal Component Analysis (PCA)

PCA is used to analyze the LIBS spectra of the gypsum samples in the present study. The spectra that were obtained from different gypsum samples are arranged in the form of matrix, which is made up with variables having 8848 features. The Unscrambler-X software (CAMO Software, Banglore, India) is used to solve the data matrix to generate PCA plot. Figure 9 illustrates that the combination of PC1 and PC2 is 92%, which is very close to 100%, which explains most of the variance that is present in the dataset. Five replicate spectra of each gypsum sample are classified in five different groups while using PCA. These groups indicate variation in elemental concentrations in gypsums. Replicates of each gypsum sample are very close to each other, forming a clustering in the score plot. Clearly, the samples of each class tend to cluster and they are well-separated from the other classes in almost all cases. The PC1 classifies gypsum samples into two groups (see Figure 9). The first group contains MAG, RDG1, and RDG2 having positive correlation. The second groups KG and MRD show negative correlation. MRD and KG are different from MAG, RDG1, and RDG2, as they contain Si, Fe, Al, Mg, and Ti. The above observation is also consistent with the loading plot, as illustrated in Figure 10. A positive correlation that was observed in the loading plot corresponds to the emission lines of Ca, which is greater in RDG1, RDG2, and MAG, while a negative correlation is noted in Si, Al, Ti, Mg, and Fe. In the second group (MRD and KG), Fe and Mn are only present in KG. The Si content is very high in MRD in comparison to KG. This is the main reason that leads to the classification of these samples into two different clusters based on PC2. Further, the cluster of KG is comparatively spread out, which suggests sample heterogeneity. Similarly, PC2 is classified into two different categories in the first group (RDG1and MAG) due to the less hydration in RGD1. The clustering of RGD2 and MAG are very close to each other, as they contain meager impurities. However, RGD2 and MAG also form two groups because of Sr and Ba that are present in RDG2.

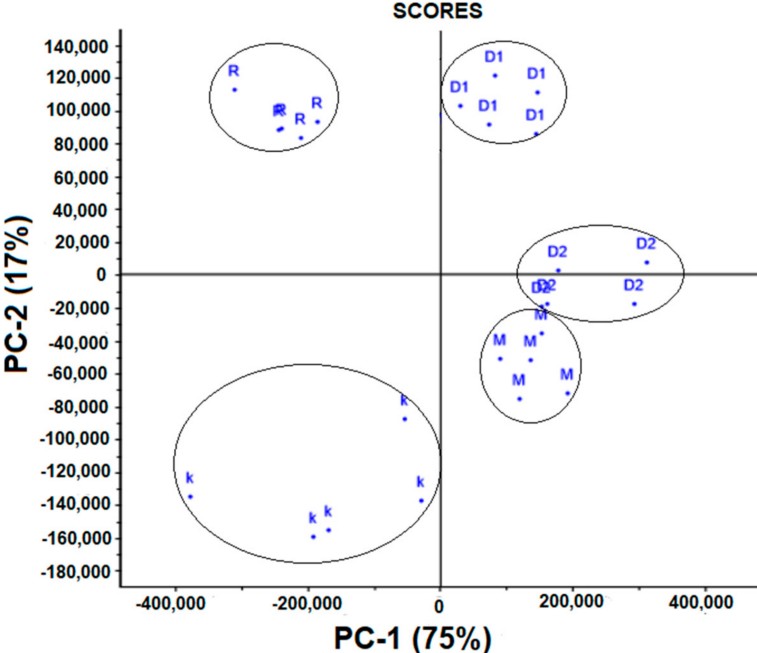

**Figure 9.** Principal component analysis (PCA) plot of the samples. The labels indicate the gypsum K = Kachchch; M = Acicular; D1 = Rajpura-Dariba 1; D2 = Rajpura Dariba 2; R = Desert Rose.

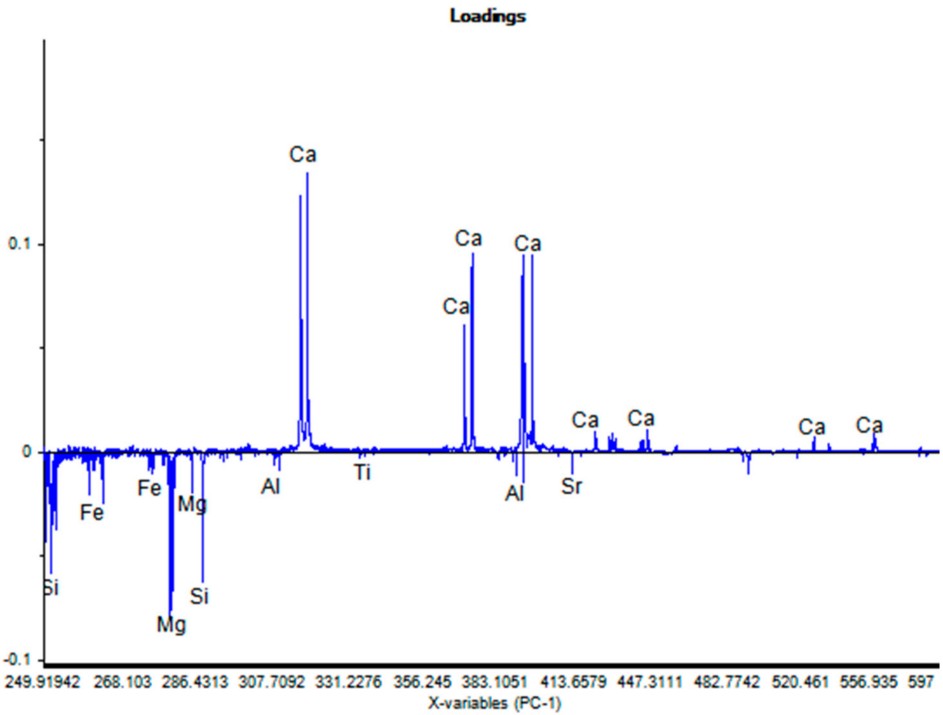

**Figure 10.** Loading plot for the principal component analysis of the LIBS spectral data.

*4.4. Partial Least-Square Regression (PLSR)*

PLSR is used to construct a model for the prediction of the elements. In this experiment, it is used to predict the concentration of main constituent of gypsum i.e., calcium and sulfur. The wavelength region having maximum number of spectral lines of corresponding elements has been included for building the calibration model. The PLSR model is employed to construct the calibration model for the gypsum samples, whose concentration was obtained from electron probe micro analysis (EPMA). The graphical results are called Predicated vs. Reference plots, see Figure 11. Similarly, the PLSR plots are generated for sulfur.

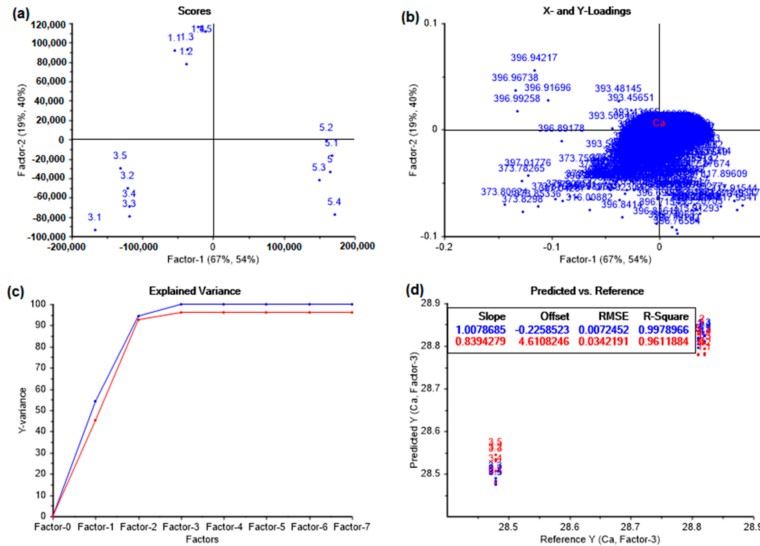

**Figure 11.** (**a**) Score plot of Partial Least-Square Regression (PLSR) model, (**b**) Loading plot of PLSR model, (**c**) Explained variance plot in PLSR model, and (**d**) Predicted vs. Reference.

The performance of the model depends upon the two parameters: (i) the coefficient of determination (R2) and (ii) root mean square error (RMSE). The model is said to be strongly correlated when the RMSE and R2 values tend to zero and one, respectively, in the plots. In this study model, the R2 values are nearly 0.96 and RMSE value is 0.02 and the calibration (blue line) and validation (red line) deviate very little from each other. Thus, the model that was adopted in the present study is strongly correlated. The concentration of all calcium and sulfur is determined while using the above PSLR model and Table 3 tabulates the values.

**Table 3.** Reference and predicted values of calcium and sulfur in analyzed gypsum samples. (a) Gypsum, Kachchh, Gujarat; (b) Gypsum, Rajpura-Dariba Cu-Pb-Zn Mine, Rajasthan; (c) Gypsum, Rajpura-Dariba Cu-Pb-Zn Mine, Rajasthan; (d) Acicular Gypsum, Morocco; and (e) Desert Rose Gypsum, Sahara Desert, Morocco.

| Sample | Calcium Concentration | | | Sulfur Concentration | | |
|---|---|---|---|---|---|---|
| | EPMA Reference Value | PLSR Predicted Value | Accuracy * | EPMA Reference Value | PLSR Predicted Value | Accuracy * |
| (a) | 24.82 | 24.115 | 101.09 | 10.76 | 11.290 | 104.92 |
| (b) | 26.45 | 25.190 | 95.23 | 13.23 | 12.113 | 91.55 |
| (c) | 25.48 | 26.878 | 105.48 | 10.91 | 10.197 | 93.46 |
| (d) | 28.99 | 28.695 | 98.98 | 14.82 | 15.406 | 103.95 |
| (e) | 23.71 | 23.41 | 98.83 | 11.23 | 10.865 | 96.74 |

$$^*\text{Accuracy} = \frac{\text{PLS predicted concentration}}{\text{Reference Concentration}} \times 100.$$

## 5. Conclusions

In this work, the terrestrial gypsum samples from varied locations and textural types have been analyzed by laser-induced breakdown spectroscopy (LIBS). Petrographic, sub-microscopic, and X-ray diffraction (XRD) investigations confirm the gypsums. The spectral lines and their respective intensities for various light (H, C, N, and O) and trace elements (Ti, Mn, Fe, Ba, and Sr) that are present in the gypsum samples are in close agreement with the results that were reported by conventional analytical techniques (INAA (Instrumental Neutron Activation Analysis), XRF (X-ray fluorescence), and LA-ICP-MS (Laser Ablation Inductively Coupled Plasma Mass Spectrometry)), moreover minor variations could also be gauged. The texturally varied gypsum samples from different locations bearing information regarding specific geological environments can be compared using LIBS data. The study highlights the LIBS sensitivity to the water content for the hydration levels of calcium sulfates. One can rapidly distinguish between anhydrite ($CaSO_4$), bassanite, $CaSO_4 \cdot 1/2H_2O$, and gypsum while using a multi-elemental analysis technique. The data collected with LIBS in this experimental study can be used as mineral references for detection of similar formation of gypsum wherever it occurs. KG gypsum contains Mn and Fe in appreciable amount indicating the availability of terrigenic sediments in the source basin.

PCA coupled LIBS shows the similarity and differences between various gypsum samples in terms of their elemental concentrations, because the samples are very well grouped into distinct clusters. The PSLR method is utilized for the prediction of the concentrations of calcium and sulfur, subsequently, this information can be used to predict the concentration of same elements in any other gypsum samples. The texturally different gypsum samples that originate from different locations can be compared, and the gypsum samples contain information on specific geological environments.

After the discovery of gypsum by the ChemCam rover, a study of Martian sediments (either in situ or in returned samples) is expected to become possible in the future. The present study demonstrated the potential application of LIBS for rapid characterization of gypsum samples, which can be extended to the study of gypsum samples in extraterrestrial environments.

**Author Contributions:** The samples are provided by J.K.P., A.K.R. conducted the experimental investigations, and all authors (A.K.R., J.K.P., C.G.P. and A.K.R.) contributed to the writing of this article.

**Funding:** This research received no external funding.

**Acknowledgments:** One of the authors, Abhishek K. Rai, expresses thanks to the UGC (CRET) of India for financial assistance of his doctoral research at the University of Allahabad, UP, India.

**Conflicts of Interest:** The authors declare no conflict of interest.

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
