# Peer review of "Plasma Spectroscopy of Various Types of Gypsum: An Ideal Terrestrial Analogue"

_atoms, doi:10.3390/atoms7030072_

Round 1

Reviewer 1 Report

In this work, gypsum samples from five different geographic locations, environments of formation and textural types have been studied using LIBS data combined with multivariate statistical analysis methods. Principal component analysis (PCA) has been used to analyze the LIBS spectra of gypsum samples, and Partial least-square regression (PLSR) has been used to construct a model for prediction of the elements. This study demonstrated the potential application of LIBS for rapid characterization of gypsum samples which can be extended to the study of gypsum samples in extraterrestrial environments.

The caption of Figure 8, must be corrected.

I recommend the publication of this work in the Journal “Atoms”.

Author Response

Dear reviewer,

We appreciate your comments.

The Figure 8 caption on page 9 has been corrected, the changes are indicated in green. Other edits on page 6 and 14 are indicated in blue.

Respectfully,

Reviewer 2 Report

In this study terrestrial gypsum samples from varied locations and textural types have been analyzed with laser-induced breakdown spectroscopy (LIBS). This research is motivated by the first detection of gypsum by the Mars Science Laboratory (MSL) rover Curiosity in the Gale Crater, demonstrating that the study of Martian sediments (either in situ or in returned samples) will become possible in the future.The present study demonstrated the potential application of LIBS for rapid characterization of gypsum samples which can be extended to the study of gypsum samples in extraterrestrial environments. The article is well written and the description of research and conclusions are clear. I have only one small remark. Acronims PCA, NIST, ICCD, EPMA, PSLR should be explained when they appear for the first time. For example, PSLR acronim is mentioned first time on the page 5 and explained on the page 6. NIST appears first time on the page 5 and explanation is on the page 7.

Author Response

Dear reviewer,

We appreciate your comments.

The acronyms have been defined on page 6 and 14. The changes are indicated in blue. Other edits are in the Figure 8 caption, indicated in green.

Respectfully,